# Immunomodulatory Effects of a Low-Molecular Weight Polysaccharide from *Enteromorpha prolifera* on RAW 264.7 Macrophages and Cyclophosphamide- Induced Immunosuppression Mouse Models

**DOI:** 10.3390/md18070340

**Published:** 2020-06-28

**Authors:** Yingjuan Liu, Xiaolin Wu, Weihua Jin, Yunliang Guo

**Affiliations:** 1Medical College, Qingdao University, Qingdao 266071, China; liuyingjuan829@163.com (Y.L.); wuxiaolin512@163.com (X.W.); 2College of Biotechnology and Bioengineering, Zhejiang University of Technology, Hangzhou 310014, China

**Keywords:** polysaccharides, *Enteromorpha prolifera*, immunomodulation, RAW 264.7 cells, cyclophosphamide

## Abstract

The water-soluble polysaccharide EP2, from *Enteromorpha prolifera*, belongs to the group of polysaccharides known as glucuronoxylorhamnan, which mainly contains glucuronic acid (GlcA), xylose (Xyl), and rhamnose (Rha). The aim of this study was to detect the immunomodulatory effects of EP2 on RAW 264.7 macrophages and cyclophosphamide (CYP)-induced immunosuppression mouse models. The cells were treated with EP2 for different time periods (0, 0.5, 1, 3, and 6 h). The results showed that EP2 promoted nitric oxide production and up-regulated the expression of pro-inflammatory cytokines, such as IL-1β, IL-6, and TNF-α, in a time-dependent manner. Furthermore, we found that EP2-activated iNOS, COX2, and NLRP3 inflammasomes, and the TLR4/MAPK/NF-κB signaling pathway played an important role. Moreover, EP2 significantly increased the body weight, spleen index, thymus index, inflammatory cell counts, and the levels of IL-1β, IL-6, and TNF-α in CYP-induced immunosuppression mouse models. These results indicate that EP2 might be a potential immunomodulatory drug and provide the scientific basis for the comprehensive utilization and evaluation of *E. prolifera* in future applications.

## 1. Introduction

Heterogeneous sulfated polysaccharides, which are ubiquitous in nature and occur in numerous organisms, possess various effective biological and pharmacological activities [1,2,3]. Recently, green algae, with the unique structural features containing major repeating disaccharide units of α-L-Rha*p*-(1→4)-D-Xyl*p* and (→4)-β-D-Glc*p*A-(1→4)-α-L-Rha*p*, have gained considerable attention for their use in functional foods and drugs due to their availability, functionalities, and low cost of production [4,5]. *Enteromorpha prolifera*, a type of green algae belonging to the phylum *Chlorophyta* and class *Chlorophyceae*, is found on seashores worldwide [6]. Previous studies revealed that polysaccharides from *E. prolifera* exert anti-oxidative, anti-microbial, and anti-hyperlipidemic effects, as well as possess the ability to improve glucose metabolism [7,8,9]. Kim et al. (2011) found that water-soluble *E. prolifera* polysaccharides produced by pyrohydrolysis and fractionation using ion-exchange chromatography could stimulate a macrophage cell line to induce the production of NO and various cytokines, but the associated mechanism was not elucidated [9]. Furthermore, no information is currently available on the effects of acidolysis-degraded polysaccharides from *E. prolifera* on macrophages. Considering that polysaccharides with a complex structure and large molecular weight were not conducive for this study, a low-molecular weight sulfated polysaccharide (EP2) was prepared and the potential immunomodulatory effects were investigated.

Macrophages derived from blood monocytes play a crucial role in host defense against infection by processing and presentation of antigens to the lymphocytes, killing pathogenic microorganisms, removing cell debris, as well as secreting pro-inflammatory mediators such as interleukin-1β (IL-1β), tumor necrosis factor-α (TNF-α) and nitric oxide (NO) [10,11]. Hence, macrophages are thought to be the important target cells of some immunomodulatory drugs. In this study, we aimed to investigate the immunomodulatory effects of EP2 on RAW 264.7 cells. Cyclophosphamide (CYP) is one of the famous anticancer agents which remains extensively used in the treatment of hematological malignancies and various epithelial tumors. However, CTX can damage the DNA of normal cells and cause immunosuppression after mainly hydrolyzed by the hepatic cytochrome P450 enzymes [12,13]. In our research, CYP was used to build an immunodeficiency model in mice, and the influence of EP2 on the immunodeficiency mice was evaluated.

## 2. Results

### 2.1. Physicochemical Properties of Polysaccharides Isolated from E. prolifera

Water-soluble polysaccharides from *E. prolifera* belonged to the group of polysaccharides known as glucuronoxylorhamnan, which mainly contains glucuronic acid (GlcA), xylose (Xyl), and rhamnose (Rha). The yield of EP1 and EP2 were 52.0% and 36.0%, respectively, and the result of chemical analysis is shown in Table 1. The major differences between EP1 and EP2 were molecular weight and the sulfate content. The sulfated contents were 13.28% and 18.99% in EP1 and EP2, respectively, and the molecular weights were 33.3 kDa and 17.3 kDa in EP1 and EP2, respectively. In addition, a marked difference was observed between EP1 and EP2 in terms of the molar ratio of Xyl to Rha (Table 2)

### 2.2. Effects of EP1-2 on Cell Viability and NO Production in RAW 264.7 Cells

To detect the immunomodulatory effects of the degraded polysaccharides, we measured cell viability and NO production after treatment with EP1 and EP2. As shown in Figure 1, EP2 exerted more marked effects on the production of NO than EP1. The bioactivity of sulfated polysaccharides depends on several structural parameters, such as the molecular weight, substitution groups and position, type of sugar, and glycosidic branching [14]. A previous study reported that sulfate content is positively correlated with superoxide radical scavenging ability [15]. From the result of this study, the sulfate content may also have some positive correlation with immune enhancement. Furthermore, EP2, due to its lower molecular weight, may have better contact with cells. These results suggest that molecular weight and sulfate content are factors that contribute to immunoregulatory activity. Therefore, we investigated the immunomodulatory effects of EP2 on RAW 264.7 macrophages and cyclophosphamide-induced immunosuppression mouse models.

As shown in Figure 1D, NO production was significantly increased by EP2 treatment in a concentration-dependent manner at concentrations from 1.25 to 40 μg/mL, and then remained constant at concentrations higher than 40 μg/mL, which showed an effect similar to that of LPS. Hence, we chose 40 μg/mL as the experimental concentration. We incubated RAW 264.7 cells with 40 μg/mL EP2 for 0, 0.5, 1, 3, and 6 h to explore the effects of EP on cytokines and inflammatory mediators over time. The polysaccharides were not contaminated with endotoxins, which indicate that the enhanced NO release was not induced by endotoxin compounds in the RAW 264.7 cells. 

### 2.3. Effects of EP2 on Cytokines and Proinflammatory Proteins in RAW 264.7 Cells

Macrophage plays a critical role in host defense me; chanism, and its activation stimulates the release of pro-inflammatory cytokines [16]. As shown in Figure 2A–D, EP2 significantly promoted the secretion of IL-1β, IL-6, and TNF-α at different time points. Furthermore, the result of immunofluorescence of TNF-α further verified the effects of EP2 on RAW 264.7 cells (Figure 2H). NO is an unorthodox messenger molecule, which can be generated by inducible nitric oxide synthase (iNOS) in macrophages [17]. To determine if the immunoregulatory effects of EP2 were involved in the suppression of NO, the expression of iNOS was measured. As shown in Figure 2E, the expression of iNOS increased in a time-dependent manner after EP2 treatment. MMP-9 and COX-2, which aggravate inflammatory response via the up-regulation of proinflammatory mediators, also increased significantly as depicted in Figure 2F,G, implying that EP2 could promote the release of cytokines to improve the immunomodulatory function of macrophages.

### 2.4. EP2 Promoted the Activation of NLRP3 Inflammasome in RAW 264.7 Cells

Pathogen-associated molecular patterns, which are mediated by TLR4, are the typical inflammatory responses of macrophages, and they could recognize exotic and activate the signal transduction system for immunoregulation [18]. As shown in Figure 3B, EP2 enhanced the expression of TLR4 in a time-dependent manner. The result of immunofluorescence test further verified the effects of EP2 on RAW 264.7 cells, indicating that EP2 could activate RAW 264.7 cells by integrating into TLR4 (Figure 3E). After pre-treatment with TLR4 inhibitor (TLR4-IN-C34) and EP2, the production of NO was significantly suppressed, but not completely (Figure 3F). These results proved that TLR4 played an important role in the immune enhancement ability of EP2, but the effect was not completely dependent on TLR4.

Nucleotide-binding oligomerization domain-like receptor family pyrin domain-containing 3 (NLRP3) inflammasome plays a crucial role in innate immune responses to pathogens or danger-associated molecular patterns, and it is a well-known inflammatory mediator which induces the overproduction of IL-1β [19]. As shown in Figure 3C,D, EP2 increased the expressions of NLRP3 and mature caspase-1, which was consistent with the immunofluorescence results presented in Figure 3F. The data suggested that EP2 promoted NLRP3 inflammasome activity by acting on the priming stage. 

### 2.5. EP2 Promoted the Activation of MAPK Signaling Pathway in RAW 264.7 Cells

MAPKs play an important role in the generation of innate immune responses and production of mediators in macrophages, including iNOS, IL-1β, and TNF-α [20]. Therefore, we examined the phosphorylation levels of MAPK signaling molecules, comprising ERK1/2, JNK, and p38 by Western blotting. As shown in Figure 4**,** the result clearly showed that EP2 significantly increased the phosphorylation of P38, ERK1/2, and JNK. However, the total protein levels of ERK1/2, JNK, and p38 were relatively constant in all the groups.

### 2.6. EP2 Promoted the Activation of NF-κB and AP-1 in RAW 264.7 Cells

TLR4 activation triggers a signaling cascade that promotes NF-κB nuclear translocation, where it acts as a transcription factor for several genes involved in inflammation [21]. As shown in Figure 5A,B, EP2 improved the phosphorylation level of inhibitor of κB (IκB α) (an inhibitor of NF-κB) which remained inactive in the cytoplasm and bound with p65. Following the inhibition of IκB α by phosphorylation, p65 was released and phosphorylated to translocate into the nucleus (Figure 5C). Another transcription factor regulating inflammatory responsive genes is activator protein-1 (AP-1), which is composed of the members of the Jun and Fos families. Activated MAPKs signaling cascades, especially the c-Jun NH2-terminal kinase (JNK) could activate AP-1, which was consistent with the enhanced protein levels of c-Jun and c-Fos induced by EP2 treatment (Figure 5D,E).

### 2.7. EP2 Improved Visceral Index and Inflammatory Cell Counts in CYP-induced Immunosuppression Mouse Models

As shown in Table 3, compared with the control group, the body weight, spleen, and thymus indices in mice after intraperitoneal injection of CYP for four days were markedly decreased. Following the intraperitoneal injection of EP2 for 14 days, the CYP-induced decrease in the indices were ameliorated (*p* < 0.05), implying that EP2 played a positive immunostimulatory role in the CYP-treated mice. Furthermore, the administration of CYP reduced the inflammatory cell counts (WBC, NEUT, LYMPH, and PLT) in mice (Table 4). However, EP2 significantly suppressed this decline. 

### 2.8. EP2 Enhanced the Production of Cytokines in CYP-induced Immunosuppression Mouse Models

The secretion of cytokines is an important indicator of the immune function in the body. To investigate the effects of EP2 on the regulation of immune function in the immunosuppressed mice, the levels of TNF-, IL-1β, and IL-6 was determined in mouse serum. As shown in Figure 6, there was a significant difference (*p* < 0.05) between the control mice and the CYP-treated mice in terms of the levels of TNF-, IL-1β, and IL-6. The secretion of the cytokines in the CYP-treated group was significantly inhibited after EP2 treatment. These results indicate that EP2 could enhance the immune function in immunocompromised mice.

## 3. Discussion

As one of the main algal genera that cause green tide, the over-growth of *E. prolifera* has a noticeable negative impact on the environment and aquaculture [9,22]. Researchers have been making great efforts to assess the value of *E. prolifera* in bio-oil extraction, marine aquaculture, food processing, and drug development. It is reported that *Enteromorpha* contains a variety of active components, including polysaccharides which are the most important components [23]. In this study, we obtained a low-molecular weight polysaccharide by degradation and anion exchange chromatography separation from *E*. *prolifera*. The polysaccharide mainly contained GlcA, Xyl, and Rha, and the molecular weight was 17.3 kDa with 18.99% sulfate content. To detect the immunomodulatory effects of EP2, we measured cell viability and NO production after RAW 264.7 cells were treated with EP2. As shown in Figure 1, EP2 significantly increased the production of NO without being harmful to RAW 264.7 cells. The presence of NO in the culture medium of macrophage cells is considered one of the most reliable factors that indicate the classical activation of macrophages [24]. 

NLRP3 inflammasome plays a crucial role in host immune responses to various pathogen-derived factors as well as danger-associated molecules, which involves the recruitment of apoptosis-associated speck-like protein containing a CARD (ASC) and caspase-1 [25,26]. NLRP3 binds to pro-caspase-1 through ASC, subsequently activating caspase-1. The activation of capase-1 leads to the maturation of IL-1β [27,28]. In this study, the expression of NLRP3 inflammasome and cleaved caspase-1 was increased by EP2 treatment, which was consistent with the increased expression of TNF-α, IL-1β, and IL-6. This suggests that EP2 can enhance immunoreaction. The pro-inflammatory cytokines such as TNF-α, IL-1β, and IL-6 may have multiple functions during immunomodulatory process. Especially, the cytokines released from immune cells can stimulate the innate immune responses, which are essential for immunomodulation [29]. 

The activation of NLRP3 inflammasome indicates the activation of NF-κB [30]. Upon stimulation with pro-inflammatory factors, IκBα was phosphorylated, selectively ubiquitinated, and then quickly degraded, which in turn leads to the liberation of NF-κB [26,31]. The liberated NF-κB dimers were transferred into the nucleus where it bound to the promoter site of the target gene to induce the transcription of pro-IL-1β and the self-assembly of NLRP3 [32]. The presence of EP2 promoted the activation of NF-κB through the enhancement of the phosphorylation of IκB α and NF-κB p65 subunit. AP-1 is another transcription factor regulating inflammatory responsive genes. It consists of the members of Jun, Fos, or ATF families. The activation of AP-1 involved the heterogeneous dimerization of Jun and Fos, and the recognition of the transcription site [33]. Treatment with EP2 induced the expression of c-Jun and c-Fos after 1 h. The activation of transcription factors promotes the synthesis and release of inflammatory factors. These results further demonstrate that EP2 stimulated the immune response of macrophages.

Recently, several lines of evidence have indicated that MAPKs, a group of downstream Ser/Thr kinases, can modulate the activities of NF-κB and AP-1 [34,35]. In response to various stimuli, the activation of JNK, ERK1/2, and p38 MAPK by phosphorylation is a key step leading to the expression of inflammatory mediators [36]. JNK and p38 strengthen the translation of TNF-α mRNA and maintain the stability of the transcriptional process. Moreover, ERK1/2 promotes the transport of TNF-α mRNA from the nucleus to the cytoplasm [37,38]. EP2 treatment promoted the phosphorylation of JNK, ERK1/2, and p38, which was supported by the increased level of TNF-α. EP2 promoted the expression of TLR4 which specifically mediates the innate immune response involved in various inflammatory disorders [39,40]. The activation of TLR4 upon EP2 stimulation sent downstream signals by secreting a variety of pro-inflammatory mediators, NO, TNF-α, and IL-1β in a time-dependent manner. The addition of TLR4 inhibitors significantly inhibited the activation of EP2 in RAW 264.7 cells, but the inhibitor did not completely block the effect of EP2. This indicates that TLR4 played an important role in the activation of EP2 in macrophages, but the activity of EP2 was not completely dependent on TLR4. After all, the mechanism of action of polysaccharides in cells is complicated.

Furthermore, we characterized the immunomodulatory effects of EP2 on immunosuppressed mice and found that these effects occurred through the regulation of immune organs, inflammatory cell counts, and cytokines. CYP as an effective immunosuppressive and chemotherapeutic agent can damage the structure of DNA, interfere with the proliferation and differentiation of macrophages, kill immune cells, and weaken the immune system of the organism [40,41]. It induces an imbalance in the immune function homeostasis. A previous study showed that CYP causes overall immunological dysfunction regardless of cell phenotypes by markedly repressing the production of cytokines [42]. Because of the broad toxicity of CYP, the CYP-induced immunosuppression model is the most commonly used in immunostimulatory experiments [43]. The administration of CYP reduced the body weight, spleen index, and thymus index. The weights are recognized as critical and intuitive indices for non-specific immunity. Thymus and spleen indices are assessed to evaluate the whole immune state of the organism [18]. In our study, EP2 significantly increased the body weight, spleen index, and thymus index, indicating that EP2 could improve the immune function of developing immune organs. Moreover, peripheral WBC, NEUT, LYMPH, and platelet counts showed that treatment with EP2 inhibited CYP-induced immunosuppression, which is an important limiting factor in the outcome and recovery of tumor patients receiving chemotherapy. This suggests that EP2 can enhance the host’s specific and non-specific immunity, including cellular and humoral immune systems. 

## 4. Materials and Methods

### 4.1. Materials

High-glucose Dulbecco’s modified Eagle’s medium (DMEM) and fetal bovine serum (FBS) were purchased from Hyclone (Logan, UT, USA). Lipopolysaccharide (LPS) (*Escherichia coli* 0111: B4) was purchased from Sigma-Aldrich (St. Louis, MO, USA). The cyclophosphamide (CYP) were obtained by Shanghai Yuanye Bio-Technology Co., Ltd. (Shanghai, China). 

### 4.2. Extraction and Purification of the Polysaccharide

*E. prolifera* (1000 g) was extracted with 0.1 M HCl (30 L) at room temperature for 4 h. Then, the supernatant was filtered through celite, neutralized, concentrated and precipitated by ethanol. The sediment (8 g, the yield was 20.1%) underwent anion exchange chromatography on a DEAE-Bio Gel Agarose FF gel (6 cm D × 40 cm H) with elution by water (5 L), 0.3 M NaCl (5 L), 1 M NaCl and 2 M NaCl, respectively. After being ultrafiltrated, concentrated, and precipitated by ethanol, the polysaccharides in 1 M NaCl (the yield was 53.2%) underwent autohydrolysis and were precipitated by ethanol according to the modified method [44]. The precipitate was fractionated on a Bio-Gel P-10 Gel column (2.6 cm D × 100 cm H) eluted with 0.5 M NH_4_HCO_3_ into two fractions (EP1 and EP2). The degraded polysaccharides (EP1 and EP2) was dialyzed against distilled water and then lyophilized.

### 4.3. Composition Analysis

Chemical compositions of polysaccharides were elucidated from the yields, total sugar content, sulfated content, uronic acid (UA) content, molar ratio of monosaccharides and the molecular weight. The content of total sugar was determined according to the modified phenol-sulfuric acid method, using rhamnose as a standard [45]. An ion chromatography with a Shodex IC SI-52 4E column (4.0 mm D × 250 mm H) was applied to perform the sulfated content with 3.6 mM Na_2_CO_3_ at a flow rate of 0.8 mL/min at 45 °C referred to the previous study [46]. UA was determined using a modified carbazole method [47], using glucuronic acid as a reference. Finally, the molar ratio of monosaccharide composition was determined according to the method described by Geng et al. [17]. Prior to HPLC, the polysaccharide was first hydrolyzed and derivatized with PMP. Briefly, polysaccharides (10 mg/mL) were hydrolysed by trifluoroacetic acid (2 M) under a nitrogen atmosphere for 4 h at 110 °C. Then, 0.2 mL of hydrolysed mixture was neutralized to pH 7 with sodium hydroxide. Later, the mixture was converted into its PMP derivatives and separated by HPLC chromatography on a YMC Pack ODS AQ column (4.6 mm D × 250 mm L). The molecular weight was analyzed by HPGPC on a TSK G3000 PWxl column (7 μm, 8.0 mm D × 300 mm H) (TOSOH, Tokyo, Japan) eluted with 0.05M Na_2_SO_4_ at a flow rate of 0.5 mL/min at 30 °C with refractive index detection. Ten different molecular weight dextrans purchased from the National Institute for the Control of Pharmaceutical and Biological Products (Beijing, China) were used as weight standards. 

### 4.4. Endotoxin Test

The concentration of endotoxin was determined by using the chromogenic end-point TAL assay kit (Solarbio, Beijing, China). The experiment was performed according to the manufacturer’s instructions. The endotoxin in the sample activates a cascade of enzymes in TAL, the activated enzyme splits the synthetic substrate, releasing a yellow colored product with maximum absorbance at 405 nm. The yellow product can further react with diazo reagents forming purple product with maximum absorbance at 545 nm. The absorbance of both the yellow product and purple product are proportional to endotoxin levels.

### 4.5. Cell Culture and Cell Viability Assay

The RAW 264.7 murine macrophages were obtained from the National Infrastructure of Cell Line Resource (Beijing, China) and cultured in DMEM high glucose medium supplemented with 100 U/mL penicillin, 100 μg/mL streptomycin and 10% heat-inactivated FBS. Cultures were maintained at 37 °C in a humidified 5% CO_2_ incubator. RAW 264.7 cells were seeded at a density of 1 × 10^5^ cells/mL in a 96-well plate overnight and then treated with various concentrations of polysaccharides or 1 μg/mL of LPS for 24 h. After 24-h incubation, CCK-8 reagent (10 µL/mL) was added to each well and the absorbance was measured at 450 nm using a microplate reader.

### 4.6. Nitric Oxide Production

RAW 264.7 cells at a density of 1 × 10^5^ cells/mL were incubated with different concentrations of polysaccharides or LPS (1 μg/mL) for 24 h. After incubation, supernatants were collected and reacted with Griess reagent according to the manufacturer’s instructions (Beyotime Biotechnology, Shanghai, China). A NaNO_2_ standard curve was used to calculate nitrite concentration.

### 4.7. Immunofluorescence Staining Analysis 

RAW 264.7 cells were grown on cover slips in six-well plates and treated with EP2 for 6 h. The cells were treated as mentioned, and then fixed and incubated with a blocking buffer for 1 h to suppress nonspecific binding. Next, cells were incubated with the primary antibody at 4 °C overnight, followed by incubation with a FITC-conjugated secondary antibody at room temperature for another 1 h. DAPI was used as nuclear staining after 3-min incubation with cells. Samples were visualized under a fluorescent inverted microscope (Olympus Fluoview FV1000, Tokyo, Japan).

### 4.8. Western Blot

RAW 264.7 cells at a density of 1 × 10^6^ cells/mL were seeded in six-well plates, then treated with EP2 for 0, 0.5, 1, 3 and 6 h. After that, cells were washed twice with phosphate buffered saline and harvested in cold lysis buffer containing protease inhibitors or phosphatase inhibitors. The proteins were collected by centrifugation. The concentration of proteins was determined by a BCA protein assay (Beyotime Biotechnology, Shanghai, China). The proteins were separated by 12.5% SDS-PAGE gels and transferred to a PVDF membrane (Millipore, Carrigtwohill, Ireland). The membrane was sealed with 5% bovine serum albumin for 2 h, then hybridized with primary antibodies (TNF-α, IL-1β, iNOS, COX-2, MMP-9, IL-6, TLR4, NLRP3, caspase-1, P38, ERK1/2, JNK, IκB, P65, c-jun, c-fos) (Abcam, Cambridge, UK) overnight at 4 °C. After being rinsed with TBST three times, the horseradish peroxidase-conjugated IgG secondary antibody (Jackson ImmunoResearch Laboratories, Inc., West Grove, PA, USA) (1:2000) was added. Enhanced chemiluminescence (Millipore, Carrigtwohill, Ireland) was applied to detect the bands of proteins.

### 4.9. Animals

The study was approved by the Animal Care and Ethics Committee of Qingdao University in compliance with the Principles of Laboratory Animal Care that were developed by the National society for Medical Research. The animals were kept with free access to food and tap water and housed under standard conditions, with controlled temperature (22 ± 2 °C) and a 12 h light–dark cycle. Before the study proceeded, fifty male ICR mice (22–25 g) were given seven days to acclimatize to the feeding environment. 

The mice were randomly divided into five groups (*n* = 10 for each group) (1) control; (2) CYP only; (3) EP2 (20 mg/kg/d) + CYP; (4) EP2 (40 mg/kg/d) + CYP; (5) EP2 (80 mg/kg/d) + CYP. The mice in groups (3–5) received different concentrations of EP2 intraperitoneally for consecutive 14 days. In addition to the control group, mice in other groups were injected intraperitoneally with CYP (50 mg/kg/d) at day 8, 10, 12 and 14. Mice were given gavage of normal saline (0.1 mL/20 g) in the control group. The experimental design was showed in Figure 7. Mice in each group were weighed from the first day of feeding. The thymus index and spleen index of each group were calculated as thymus and spleen weight.

### 4.10. Inflammatory Cell Counts

Twenty-four hours after the last drug administration, whole blood samples were collected into heparin tubes. The total number of red blood cell (RBC), peripheral white blood cell (WBC), neutrophil (NEUT), lymphocyte (LYMPH) and platelet (PLT) were counted using a SYSMEX XS-500i Analyzer (Kobe, Japan).

### 4.11. Enzyme Linked Immunosorbent Assay (ELISA)

TNF-α and IL-1β were measured by ELISA kit. Briefly, the blood samples were collected by the heart punctures in mice, then centrifuged at 1200 g for 10 min to obtain serum. The levels of TNF-α, IL-6 and IL-1β in the serum were detected according to the instruction (Multi Sciences (Lianke) Biotech, Co., Ltd, Hangzhou, China).

### 4.12. Statistical Analysis

The data and statistical analyses comply with the recommendations on experimental design and analysis. Data analysis was performed using 19.0 SPSS software (SPSS Inc., Chicago, IL, USA). One-way analysis of variance (ANOVA) was used to compare the mean differences among the groups. The results are shown as the mean ± SEM and a level of *p* ≤ 0.05 or *p* ≤ 0.01 was considered to be statistically significant.

## 5. Conclusions

In conclusion, our study demonstrated that EP2 isolated from *E. prolifera* acted as an immunomodulatory mediator by releasing cytokines, such as IL-1β, TNF-α and IL-6, as well as activating iNOS, COX-2, and NLRP3 inflammasomes in RAW 264.7 macrophages, in which the TLR4/MAPK/NF-κB signaling pathway played an important role. Moreover, EP2 significantly increased the body weight, spleen index, and thymus index, and raised the inflammatory cell counts in CYP-induced immunosuppression mouse models. Our findings provide insights into the immunomodulatory function of EP2 and the scientific basis for the comprehensive utilization and evaluation of *E. prolifera* in future applications.

## Figures and Tables

**Figure 1 marinedrugs-18-00340-f001:**
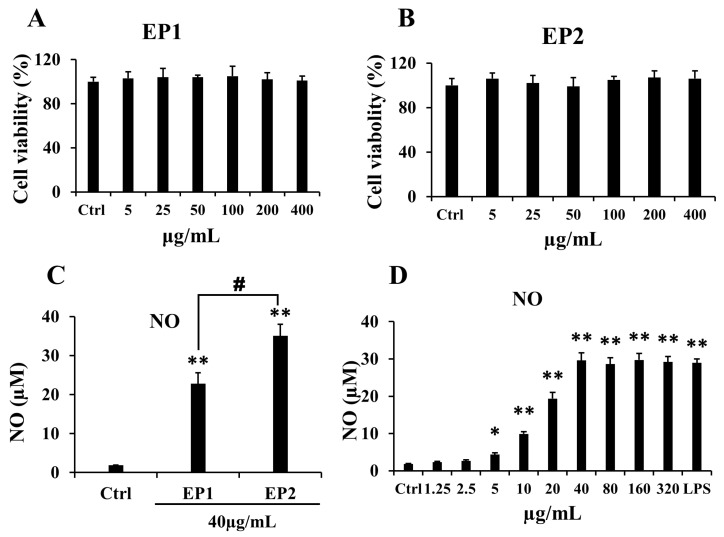
Effects of EP2 on cell viability and NO production in RAW 264.7 cells. The RAW 264.7 cells were incubated with different concentrations of EP1 or EP2 for 24 h, LPS (1 μg/mL) as a positive control. Cell viability of EP1 (**A**) and EP2 (**B**); (**C**) NO production of EP1 and EP2 at 40 μg/mg. (**D**) NO production of different concentrations of EP1. Data were expressed as mean ± SD (*n* = 6). Significant difference from the control group was designated as * *p* ≤ 0.05 and ** *p* ≤ 0.01. ^#^
*p* ≤ 0.05 showed significant difference between EP1 and EP2.

**Figure 2 marinedrugs-18-00340-f002:**
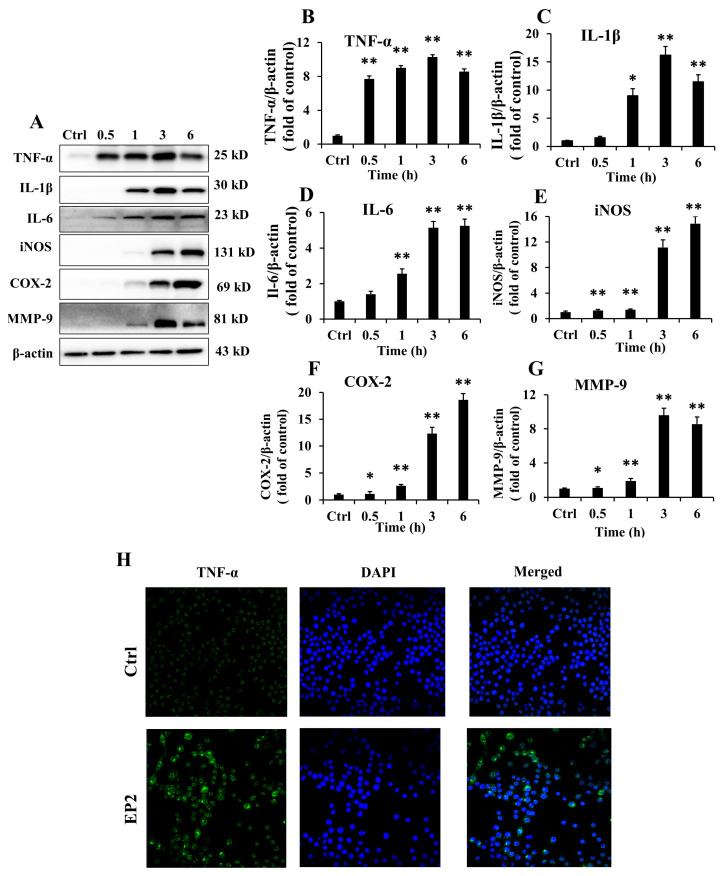
Effects of EP2 on cytokines and proinflammatory proteins in RAW 264.7 cell. Cells were incubated with 40 µg/mL EP2 for 0, 0.5, 1, 3 and 6 h. (**A**) The original bands of TNF-α, IL-6, IL-1β, iNOS, COX2, MMP-9 and β-actin; (**B**) Quantification of TNF-α; (**C**) Quantification of IL-1β; (**D**) Quantification of IL-6; (**E**) Quantification of iNOS; (**F**) Quantification of COX2; (**G**) Quantification of MMP-9; (**H**) Immunostaining of TNF-α. Data were expressed as mean ± SD (*n* = 3). Significant difference from the control group was designated as * *p* < 0.05 and ** *p* < 0.01.

**Figure 3 marinedrugs-18-00340-f003:**
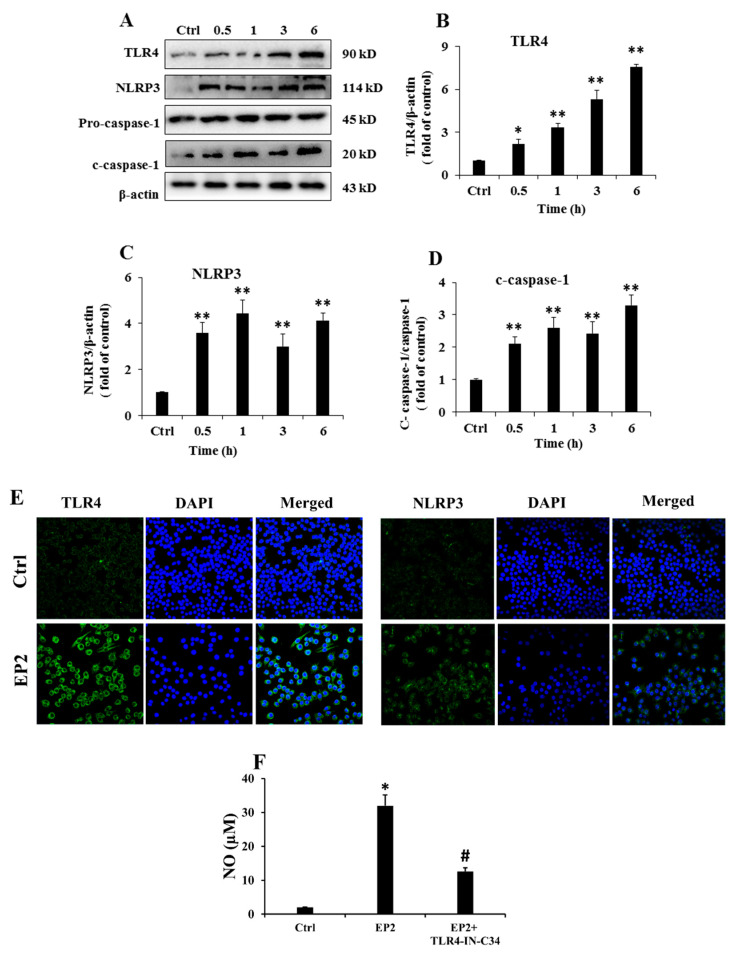
EP2 promoted the activation of NLRP3 inflammasome in RAW 264.7 cell. Cells were incubated with 40 µg/mL EP2 for 0, 0.5, 1, 3 and 6 h. (**A**) The original bands of TLR4, NLRP3, caspase-1 and β-actin; (**B**) Quantification of TLR4; (**C**) Quantification of NLRP3; (**D**) Quantification of caspase-1; (**E**) Immunostaining of TLR4 and NLRP3. The RAW 264.7 cells were treated with TLR4 inhibitor (TLR4-IN-C34) for 60 min followed by treatment with EP2 for 24 h; NO content in supernatant was determined by Griess reagent (**F**). Data were expressed as mean ± SD (*n* = 3). Significant difference from the control group was designated as * *p* < 0.05 and ** *p* < 0.01. Significant difference between EP2 group and EP2+TLR4-IN-C34 group was designated as ^#^ p < 0.01.

**Figure 4 marinedrugs-18-00340-f004:**
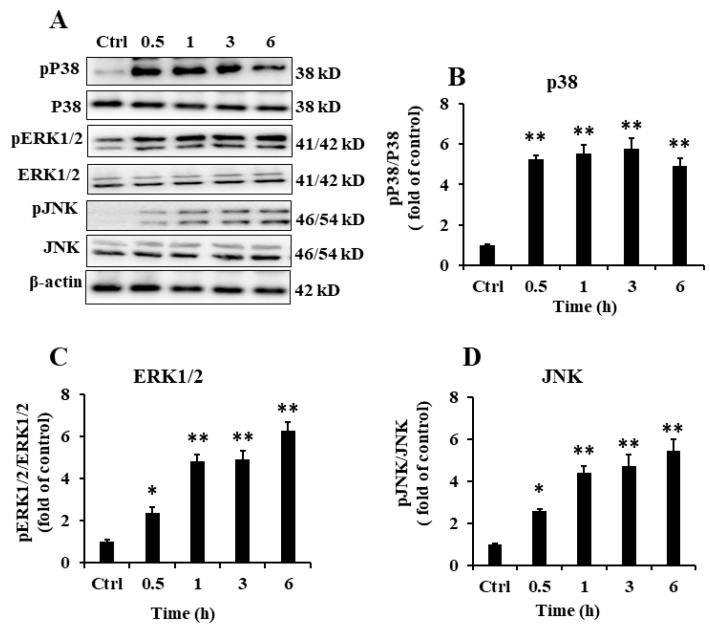
Effect of EP2 on the expression of MAPK signal pathway in RAW 264.7 cells. Cells were incubated with 40 µg/mL EP2 for 0, 0.5, 1, 3 and 6 h. (**A**) The original bands of p-P38, P38, p-ERK, ERK, p-JNK, JNK and β-actin; (**B**) Quantification of P38; (**C**) Quantification of ERK; (**D**) Quantification of JNK. Data were expressed as mean ± SD (*n* = 3). Significant difference from the control group was designated as * *p* < 0.05 and ** *p* < 0.01.

**Figure 5 marinedrugs-18-00340-f005:**
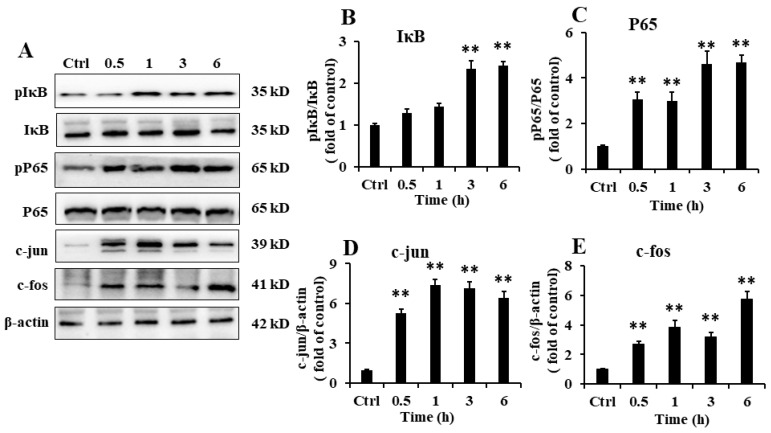
Effects of EP2 on the expression of NF-κB and AP-1 in RAW 264.7 cells. Cells were incubated with 40 µg/mL EP2 for 0, 0.5, 1, 3 and 6 h. (**A**) The original bands of p-IKB α, IKB α, p-P65, P65, c-jun, c-fos and β-actin; (**B**) Quantification of IKB α; (**C**) Quantification of P65; (**D**) Quantification of c-jun; (**E**) Quantification of c-fos. Data were expressed as mean ± SD (*n* = 3). Significant difference from the control group was designated as ** *p* < 0.01.

**Figure 6 marinedrugs-18-00340-f006:**
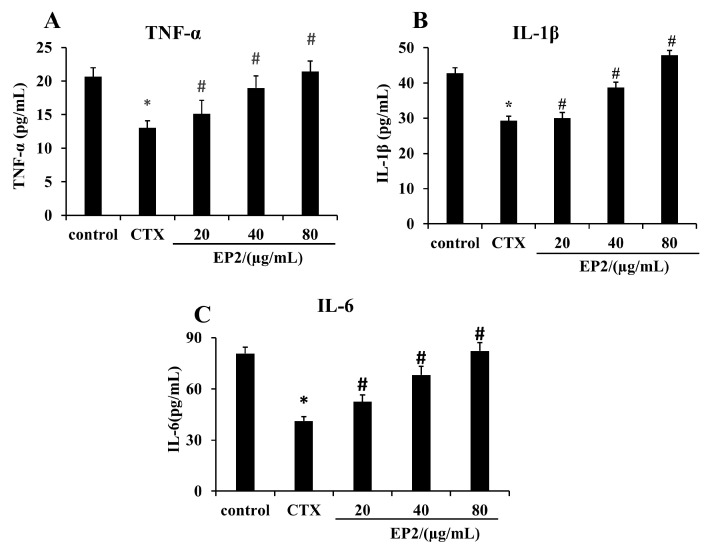
EP2 enhanced TNF-α, IL-1β and IL-6 in CYP-induced immunosuppressed mice. The protein levels of TNF-α, IL-1β and IL-6 in serum were measured by ELISA. The levels of TNF-α (**A**), IL-1β (**B**) and IL-6 (**C**) measured by ELISA. Data were expressed as mean ± SD (*n* = 3). Significant difference from the control group was designated as * *p* < 0.05. Significant difference from the CYP group was designated as ^#^
*p* < 0.05.

**Figure 7 marinedrugs-18-00340-f007:**
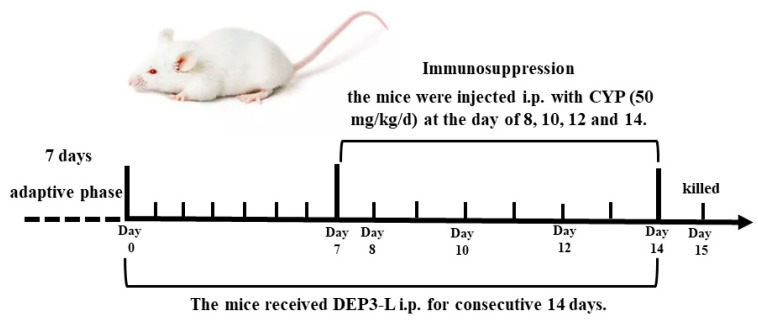
**Fig****ure****7****.** The experimental design of EP2 alleviated CYP-induced immunosuppression in ICR mice.

**Table 1 marinedrugs-18-00340-t001:** Chemical compositions of the polysaccharides studied.

Samples	Yields (%)	UA (%)	SO4^2−^ (%)	Total Sugar (%)	Mw (kDa)
EP1	52.0	17.3	13.3	76.8	8
EP2	36.0	15.4	19.0	82.8	4

**Table 2 marinedrugs-18-00340-t002:** Monosaccharides of the polysaccharides studied.

Samples	Monosaccharides (Molar Ratio)					
	Man	Rha	Glc A	Glc	Gal	Xyl
EP1	-	1	0.29	-	0.07	0.27
EP2	-	1	0.2	-	0.01	0.56

**Table 3 marinedrugs-18-00340-t003:** Determination of visceral index.

Sample	Body Weight (g)	Spleen Index (mg/g)	Thymus Index (mg/g)
control	30.12 ± 1.28	4.08 ± 0.42	2.16 ± 0.21
CYP	25.03 ± 1.76 *****	2.91 ± 0.61 *****	1.95 ± 0.38
EP2 (20 mg/kg/d)	27.62 ± 1.40	3.31 ± 0.84	2.02 ± 0.41
EP2 (40 mg/kg/d)	29.64 ± 1.21 **^#^**	4.11 ± 0.49 **^#^**	2.13 ± 0.57
EP2 (80 mg/kg/d)	29.22 ± 1.65 **^#^**	4.36 ± 0.47 **^#^**	2.21 ± 0.63

Note: Data were expressed as mean ± SD (*n* = 5). Significant difference from the control group was designated as * *p* < 0.05. Significant difference from the CYP group was designated as ^#^
*p* < 0.05.

**Table 4 marinedrugs-18-00340-t004:** Effects of EP2 on inflammatory cell counts in CYP-induced immunosuppressed mice.

Sample	WBC (10^9^/L)	NEUT (10^9^/L)	LYMPH (10^9^/L)	RBC (10^9^/L)	PLT (10^11^/L)
control	4.24 ± 0.58	0.24 ± 0.02	3.68 ± 0.42	7.65 ± 0.82	8.62 ± 0.75
CYP	1.96 ± 0.43 *****	0.11 ± 0.01 *****	0.97 ± 0.21 *****	6.03 ± 0.64	5.76 ± 0.82 *****
EP2 (20 mg/kg/d)	2.87 ± 0.71 ^#^	0.16 ± 0.03	2.14 ± 0.45 ^#^	7.42 ± 0.81	7.02 ± 0.54 ^#^
EP2 (40 mg/kg/d)	4.09 ± 0.45 ^#^	0.19 ± 0.03 ^#^	3.47 ± 0.63 ^#^	7.36 ± 0.65	7.63 ± 0.61 ^#^
EP2 (80 mg/kg/d)	5.11 ± 0.72 ^#^	0.22 ± 0.02 ^#^	3.82 ± 0.66 ^#^	7.67 ± 0.73	8.24 ± 0.68 ^#^

Note: Data were expressed as mean ± SD (*n* = 5). Significant difference from the control group was designated as * *p* < 0.05. Significant difference from the CYP group was designated as ^#^
*p* < 0.05.

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
