# Peer review of "Immunomodulatory Effects of a Low-Molecular Weight Polysaccharide from Enteromorpha prolifera on RAW 264.7 Macrophages and Cyclophosphamide- Induced Immunosuppression Mouse Models"

_marinedrugs, 2020, doi:10.3390/md18070340_

Round 1

Reviewer 1 Report

Page

Line

Comment

My background and interest is in the analytical chemistry used to characterise the material. Without good characterisation the study is not repeatable - increasingly recognised that providing enough information is fundamental to credible publication.

2

45-46

The purpose of the study depends on good analysis of the LMW material: required for reproducibility. This is not produced. There is no information about residual protein quantities or identities, or about residual nucleic acids. Table 1 reports, for example, EP1 as containing 76.33% sugar and 13.28% saccharide - about 90% of the total. What were the residue? Moisture? Protein? Other reagents?

2

59-60

Talks about yield, but does not say what is the baseline material - algal dry weight? Total extracted polysaccharide? Or something else?

2

Table 1

These are unrealistic numbers: citing values to ± 0.01% or ± 100 Da is inappropriate, especially when using non-commutable colorimetric methods. Authors should be aware of the uncertainty on their measurements, from analytical method validation, and cite uncertainty in their results. The number of significant figures in the values cited should be consistent with the uncertainty.

2

Table 1

The molecular weights reported are inconsistent with the reported [Materials and Methods section] fractionation on a P-10 column, which has a fractionation range of 1.5 to 10 kDa for dextrans.

2

Table 2

Earlier text [lines 33-34] described two major disaccharide repeats, which suggests that the sum of proportions [GlcA + Xyl] should equal that of Rha. The numbers, especially for EP1 are very significantly different [0.56 instead of 1]. Could the authors explain this discrepancy?

Tables 1 and 2

Calculating the figures for sugar content and composition in Tables 1 and 2 indicate that they are inconsistent: for example, the values for composition of EP1 in Table 2 indicate a value of UA% in Table 1 of only 13%. The authors should ensure that analytical values are consistent.

2

70-72

"The bioactivity……and glycosidic branching [12]". The authors of this paper provide no information on glycosidic branching for the material that they tested. This suggests that more rigorous analysis is required.

3

Figure 1

This requires better labelling of the x-axis to include units.

Sections 2.2 to 3.3

Why do the Section numbers jump, so that Section 3.3 follows 2.2? It seems poor experimental design that there is no overlap between the 24 hr time scale in Section 2.2 and the 0 to 6 hr time scale in Section 3.3

4

Figure 2

The Materials and Methods Section reports use of ELISA [Section 4.11] to quantify TNF-a and IL-1b, but only [Section 4.8] Western Blot to DETECT the other proteins. Figure 2 provides quantitative estimates of these proteins. How was detection converted to quantitation? If different quantitation methods were used for different proteins, this should been mentioned in the Figure Legend. If Western Blot was used for quantification, then full details [instrumentation, standards reagents, uncertainty of measurement etc.] should be reported.

10

278-281

Polysaccharide was fractionated on BioGel P-10 to give fractions of MW 17.3 and 33.3 kDa. The cited fractionation range for this gel is 1.5 to 10 kDa. There is clearly an inconsistency here which the authors should resolve.

10

295-298

Molecular weight estimated by HPGPC column TSK G3000 PWXL column. No information on how the column was calibrated. No information on uncertainty of measurement. Are the authors aware that sulphated polysaccharides "run high" with respect to molecular weight as calibrated using uncharged polysaccharide calibrants [if that was how they calibrated their column]?

11

316-319

This cites the use of "the manufacturer's instructions" for the Griess reagent, but no information has been provided about the supplier.

Author Response

Dear Reviewer:

On behalf of my co-authors, we thank you very much for giving us an opportunity to revise our manuscript, we are very grateful to your comments for the manuscript in a timely and professional manner. We appreciate the concerns and suggestions provided, and have revised our manuscript accordingly. At this time, we have resubmitted the revised manuscript, and hope to have an opportunity to publish this paper in Marine Drugs. The revised manuscript with corrected sections marked in red is attached for easier editing and review purposes. The questions were answered below. We would like to express our great appreciation to you for comments on our paper. Looking forward to hearing from you.

Thank you and best regards. 

Yours sincerely,

Below, please find our point-by-point responses to the comments of reviewers.

Comment 1: Page 2 Line 45-46

The purpose of the study depends on good analysis of the LMW material: required for reproducibility. This is not produced. There is no information about residual protein quantities or identities, or about residual nucleic acids. Table 1 reports, for example, EP1 as containing 76.33% sugar and 13.28% saccharide - about 90% of the total. What were the residue? Moisture? Protein? Other reagents?

Response 1: We are fully agreed with the reviewer’s comments. Good analysis is needed for researches. However, we are focused on the sulfated polysaccharides in this study. The major compositions of the sulfated polysaccharides were total sugar and sulfate. There might be some proteins, ash content, moisture or others. In addition, the total sugar was determined by phenol sulfuric acid assay using rhamnose as a standard. Different monosaccharide had different absorbance in the same concentration, suggesting that there might be some errors for the total sugar analysis.

Comment 2: Page 2 Line 59-60

Talks about yield, but does not say what is the baseline material-algal dry weight? Total extracted polysaccharide? Or something else?

Response 2: Thank you. E. prolifera (1000 g) was extracted with 0.1 M HCl (30 L) at room temperature for 4 h. And then the supernatant was filtered through celite, neutralized, concentrated and precipitated by ethanol. The sediment (8 g, the yield was 20.1%) underwent anion exchange chromatography on a DEAE-Bio Gel Agarose FF gel (6 cm ×40 cm) with elution by water (5L), 0.3 M NaCl (5 L), 1M NaCl and 2M NaCl, respectively. After ultrafiltrated, concentrated, and precipitated by ethanol, the polysaccharides in 1 M NaCl was autohydrolysis and precipitated by ethanol. The precipitate was fractionated on a Bio-Gel P-10 Gel column (2.6 × 100 cm) eluted with 0.5 M NH4HCO3 into two fractions (EP1 and EP2).

The total yields were calculated from the equation: the weight of EP1 or EP2/ the weight of polysaccharides in 1 M NaCl.

Comment 3: Page 2 Table 1

These are unrealistic numbers: citing values to ± 0.01% or ± 100 Da is inappropriate, especially when using non-commutable colorimetric methods. Authors should be aware of the uncertainty on their measurements, from analytical method validation, and cite uncertainty in their results. The number of significant figures in the values cited should be consistent with the uncertainty.

Response 3: Thank you. We have revised.

Comment 4: Page 2 Table 1

The molecular weights reported are inconsistent with the reported [Materials and Methods section] fractionation on a P-10 column, which has a fractionation range of 1.5 to 10 kDa for dextrans.

Response 4: Thank you for pointing out this error, which is corrected in our revised paper. The GPC-HPLC spectra of DEP3-H (EP1) and DEP3-L (EP2) were attached as follows.

Comment 5: Page 2 Table 2

Earlier text [lines 33-34] described two major disaccharide repeats, which suggests that the sum of proportions [GlcA + Xyl] should equal that of Rha. The numbers, especially for EP1 are very significantly different [0.56 instead of 1]. Could the authors explain this discrepancy?

Response 5: Polysaccharides from E. prolifera not only contained disaccharide repeats but also had a type of rhamnan-type sulfated polysaccharides (J.F. Cui, H. Ye, Y.J. Zhu, Y.P. Li, J.F. Wang,P. Wang, Characterization and Hypoglycemic Activity of a Rhamnan-Type Sulfated Polysaccharide Derivative, Mar Drugs, 17 (2019) ).

Comment 6: Table 1 and Table 2

Calculating the figures for sugar content and composition in Tables 1 and 2 indicate that they are inconsistent: for example, the values for composition of EP1 in Table 2 indicate a value of UA% in Table 1 of only 13%. The authors should ensure that analytical values are consistent.

Response 6: Uronic acid was determined by carbazole method using glucuronic acid as a reference. And the molar ratio of monosaccharide analysis was determined by the monosaccharide-PMP derivates HPLC method. The reaction conditions of these two methods were different, leading to the deviation.

Comment 7: Page 2 Line 70-72

The bioactivity……and glycosidic branching [12]". The authors of this paper provide no information on glycosidic branching for the material that they tested. This suggests that more rigorous analysis is required.

Response 7: We are fully agreed with the reviewer’s comments. Further studies on the structural analysis are under going, including the preparation of oligomers.

Comment 8: Page 3 Figure 1

This requires better labelling of the x-axis to include units.

Response 8: We have added the labeling of the x-axis.

Comment 9: Sections 2.2 to 3.3

Why do the Section numbers jump, so that Section 3.3 follows 2.2? It seems poor experimental design that there is no overlap between the 24 h time scale in Section 2.2 and the 0 to 6 hr time scale in Section 3.3

Response 9: Thank you for pointing out this error. I am sorry that the Results section showed wrong serial numbers, and the right numbers have been added marked in Red. To detect the immunomodulatory effects of the degraded polysaccharides, we measured cell viability and NO production after treatment with EP2. The RAW 264.7 murine macrophages were seeded at a density of 1 × 105 cells/mL in a 96-well plate and then treated with various concentrations of polysaccharides for 24 h. After 24 h incubation, CCK-8 reagent was added to each well and the absorbance was measured at 450 nm. Furthermore, the cells were treated as above and the supernatants were collected and reacted with Griess reagent to detect nitrite concentration. As shown in Fig. 1, EP2 exerted significant effects on the production of NO. Then we treated RAW 264.7 cells with EP2 for different times (0 h, 0.5 h, 1 h, 3 h, 6 h, 12 h, and 24 h) to measure the changes of EP2 on RAW 264.7 cells. And we found that the levels of many cytokines and proinflammatory proteins, such as TNF-α, IL-1β, COX-2 and iNOS were low or flat after the cells were treated with EP2 for 6 h [This is a preliminary experiment]. So, we chose the culture times (0h, 0.5h, 1h, 3h, and 6h) to detect the immunomodulatory changes of EP2 on RAW 264.7 cells.

Comment 10: Page 4 Figure 2 The Materials and Methods Section reports use of ELISA [Section 4.11] to quantify TNF-a and IL-1b, but only [Section 4.8] Western Blot to DETECT the other proteins. Figure 2 provides quantitative estimates of these proteins. How was detection converted to quantitation? If different quantitation methods were used for different proteins, this should been mentioned in the Figure Legend. If Western Blot was used for quantification, then full details [instrumentation, standards reagents, uncertainty of measurement etc.] should be reported.

Response 10: We are fully agreed with the reviewer’s suggestion. The levels of TNF-α and IL-1β in the blood samples collected by the heart punctures in mice were measured by ELISA kit. And the expressions of TNF-α and IL-1β in RAW 264.7 cells were measured by western blot. The cells were cultured with EP2 for 0 h, 0.5 h, 1 h, 3 h, and 6 h. The culture time of EP2 and RAW 264.7 cells was too short, the accumulation of cytokines in the medium was too low to show the change of cytokines with time. Western blot could detect the changes in cytokines within the cell, which were more sensitive to capture the changes. And the details of Western Blot were added.

Comment 11: Page 10 Line 278-281 Polysaccharide was fractionated on BioGel P-10 to give fractions of MW 17.3 and 33.3 kDa. The cited fractionation range for this gel is 1.5 to 10 kDa. There is clearly an inconsistency here which the authors should resolve.

Response 11: Thank you for pointing out this error, which is corrected in our revised manuscript. The GPC-HPLC spectra of DEP3-H (EP1) and DEP3-L (EP2) were attached as follows.

Comment 12: Page 10 Line 295-298 Molecular weight estimated by HPGPC column TSK G3000 PWXL column. No information on how the column was calibrated. No information on uncertainty of measurement. Are the authors aware that sulphated polysaccharides "run high" with respect to molecular weight as calibrated using uncharged polysaccharide calibrants [if that was how they calibrated their column]?

Response 12: Thank you. There was no sulfated polysaccharides standard. Therefore, ten different molecular weight dextrans purchased from the National Institute for the Control of Pharmaceutical and Biological Products (Beijing, China) were used as weight standards. 

Comment 13: Page 11 Line 316-319 This cites the use of "the manufacturer's instructions" for the Griess reagent, but no information has been provided about the supplier.

Response 13: Thank you for your advice. The information of the manufacturer's instructions for the Griess reagent has been added marked in Red.

Reviewer 2 Report

This manuscript entitled “Immunomodulatory effects of a low-molecular weight polysaccharide from Eenteromorpha prolifera on RAW264.7 macrophages and cyclophosphamide-induced immunosuppression mouse models” by Liu et al., demonstrated that EP2 from E. prolifera activated mouse macrophage RAW264.7 cells to induce pro-inflammatory cytokines and NO production. In addition, EP2 moderated cyclophosphamide-induced immunosuppression in mice. These results suggest that EP2 is a promising drug for enhancing immune systems. I would like to recommend this manuscript to be acceptable in Marine Drugs after major revision. I wrote the comments below.

Major comments

In introduction, it may be a good idea to write the reason why you carried out cyclophosphamide-induced immunosuppression mouse models. Actually, patients with anti-cancer drugs have immunosuppression. So, EP2 may be able to improve immune systems in immunosuppressed patients receiving anti-cancer drugs.

At Table 1, EP2 has sulfate group. Can you guess which monosaccharides does sulfate group mainly be modified to? Because sulfate group is very important for biological activities of polysaccharides. Furthermore, if sulfate group of EP2 are decreased and/or removed by using chemical and/or enzymatic reaction, can de-sulfated EP2 still activate RAW264.7 cells?

Protein expression levels of NLRP3 at Fig. 3A, and c-jun and c-fos at Fig. 5A rapidly increased at only 0.5 h incubation. Is this right? Did you check mRNA levels of these proteins in RAW264.7 cells?

In Fig. 3F, TLR4-IN-C34 inhibited NO production from EP2-teated RAW264.7 cells. Is TLR4 a receptor for EP2? In addition, the inhibitor did not inhibit NO production completely. Are there any other potential receptors for EP2?

At Fig. 4, EP2 rapidly induced phosphorylation of P38 compared with ERK1/2 and JNK. How do you think about it? In addition, can the inhibitors on these MAP kinases decrease pro-inflammatory cytokines and NO production?

Did you check toxicity of EP2 on ICR mice? In Table 3, spleen index of EP2 80 mg/kg is heavier than that of CYP group, indicating that EP2 administration may cause enlarged spleen. Furthermore, EP2 stimulates RAW264.7 cells to produce pro-inflammatory cytokines and NO. Overproduction of these immunomodulatory substances may cause endotoxin-like shock in mice.

Minor comments

P1, L21, AMPK should be changed to MAPK.

At Fig. 2A, presentation (Ctrl, 0.5…6) is not corresponding to bands, respectively. Correct it, please.

P11, L328, RAW274.7 should be changed to RAW264.7.

Author Response

Dear Reviewer,

Thank you very much for your evaluation and comments on our paper. We have revised the manuscript according to your kind advices and detailed suggestions. And we Enclosed please find the responses to you. We sincerely hope this manuscript will be finally acceptable to be published on Marine Drugs. Thank you very much for all your help and looking forward to hearing from you soon. We would like to express our great appreciation to you for comments on our paper.

Thank you and best regards. 

Yours sincerely,

Below, please find our point-by-point responses to the comments.

Comment 1: In introduction, it may be a good idea to write the reason why you carried out cyclophosphamide-induced immunosuppression mouse models. Actually, patients with anti-cancer drugs have immunosuppression. So, EP2 may be able to improve immune systems in immunosuppressed patients receiving anti-cancer drugs.

Response 1: Thank you for your suggestion. Cyclophosphamide (CYP) is one of the famous anticancer agents which remains extensively used in the treatment of haematological malignancies and various epithelial tumors. However, CTX can damage the DNA of normal cells and cause immunosuppression after mainly hydrolyzed by the hepatic cytochrome P450 enzymes. In our research, CYP was used to build an immunodeficiency model in mice, and the influence of EP2 on the immunodeficiency mice was evaluated. The reason carrying out cyclophosphamide-induced immunosuppression mouse models was added marked in Red.

Comment 2: At Table 1, EP2 has sulfate group. Can you guess which monosaccharides does sulfate group mainly be modified to? Because sulfate group is very important for biological activities of polysaccharides. Furthermore, if sulfate group of EP2 are decreased and/or removed by using chemical and/or enzymatic reaction, can de-sulfated EP2 still activate RAW264.7 cells?

 Response 2: Thank you. According to the previous studies ([1] J.T. Kidgell, M. Magnusson, R. de Nys,C.R.K. Glasson, Ulvan: A systematic review of extraction, composition and function, Algal Research, 39 (2019) 101422. [2] Y. Yu, Y. Li, C. Du, H. Mou,P. Wang, Compositional and structural characteristics of sulfated polysaccharide from Enteromorpha prolifera, Carbohydr Polym, 165 (2017) 221-228. [3] Y. Li, W. Li, G. Zhang, X. Lu, H. Hwang, W.G. Aker, H. Guan,P. Wang, Purification and characterization of polysaccharides degradases produced by Alteromonas sp. A321, Int J Biol Macromol, 86 (2016) 96-104. [4] M. Lahaye,A. Robic, Structure and Functional Properties of Ulvan, a Polysaccharide from Green Seaweeds, Biomacromolecules, 8 (2007) 1765-1774.), sulfation might be substituted at C3 of Rha and C2 of Xyl. In our preliminary experiment, we detected the immunomodulatory effects of EP2 and its degradations—DEP2, and we found DEP2 showed little immune-promoting effects on RAW 264.7 cells. DEP2 were the degradations of EP2, which were a series of mixtures of tetrasaccharides and hexosaccharides without sulfate groups. So, we speculated that the sulfate group played an important role in the immune-enhancement of EP2.

Comment 3: Protein expression levels of NLRP3 at Fig. 3A, and c-jun and c-fos at Fig. 5A rapidly increased at only 0.5 h incubation. Is this right? Did you check mRNA levels of these proteins in RAW264.7 cells?

Response 3: Thank you. The levels of protein expression in our research are the result of repeated experiments. When macrophages are stimulated by EP2, the nuclear proteins c-jun and c-fos will react quickly and accelerate the synthesis of pro-inflammatory cytokines and related proteins to improve the immune enhancement of the cells. Therefore, it is correct that the expression of c-jun and c-fos increases rapidly after 0.5 h incubation with EP2. NLRP3 inflammasome plays a crucial role in innate immune responses to pathogens or danger-associated molecular patterns, and it is a well-known inflammatory mediator which induces the overproduction of IL-1β. The level of NLRP3 rapidly rising after 0.5 h incubation with EP2. And the level of IL-1β was improved significantly after 1 h incubation with EP2. The expression of NLRP3 inflammasome was consistent with the increased expression of IL-1β. So, it is right that the protein expression levels of NLRP3 at Fig. 3A and c-jun and c-fos at Fig. 5A rapidly increased at only 0.5 h incubation.

   Because the final changes in the cells are reflected in the changes in the expression of the protein—the executor of the cell, we didn’t check mRNA levels of these proteins in RAW 264.7 cells. And thank you for your advice, we will detect the changes in cells by combining proteins and mRNA in my future experiments.

Comment 4: In Fig. 3F, TLR4-IN-C34 inhibited NO production from EP2-teated RAW264.7 cells. Is TLR4 a receptor for EP2? In addition, the inhibitor did not inhibit NO production completely. Are there any other potential receptors for EP2?

 Response 4: Pathogen-associated molecular patterns, which are mediated by TLR4, are the typical inflammatory responses of macrophages, and they could recognize exotic and activate the signal transduction system for immunoregulation. As shown in Fig. 3B, EP2 enhanced the expression of TLR4 in a time-dependent manner. The result of immunofluorescence test further verified the effects of EP2 on RAW 264.7 cells, indicating that EP2 could activate RAW 264.7 cells by integrating into TLR4 (Fig.3E). After pre-treatment with TLR4 inhibitor (TLR4-IN-C34) and EP2, the production of NO was significantly suppressed, but not completely (Fig. 3F). These results proved that TLR4 played an important role in the immune enhancement ability of EP2, but the effect was not completely dependent on TLR4. So, we speculated that TLR4 might be one receptor for EP2. Furthermore, there may be other receptors or other ways for EP2 to influence the inflammatory response of cells. And we will focus our attention on the immunomodulatory mechanism of EP2, including the receptors for EP2.

Comment 5: At Fig. 4, EP2 rapidly induced phosphorylation of P38 compared with ERK1/2 and JNK. How do you think about it? In addition, can the inhibitors on these MAP kinases decrease pro-inflammatory cytokines and NO production?

 Response 5: MAPK cascade activation is the center of many signaling pathways. It is a kind of important molecule that receives the signal of membrane receptor conversion and transfer and brings it into the nucleus. When cells are stimulated by growth factors or other factors, MAPKs are activated through receiving activation signals from MKK and MKKK, and show progressive phosphorylation. Among mammals, ERK is widely found in various tissues and participates in the regulation of cell proliferation and differentiation. A variety of growth factor receptors, nutrition-related factor receptors need ERK activation to complete the signal transduction process. JNK family is the key molecule of cell signal transduction induced by various stressors and is involved in the stress response of cells to radiation, osmotic pressure, temperature change and so on. p38 mediated inflammation, apoptosis, thus become a target for the development of anti-inflammatory drugs. Thus, EP2 rapidly induced phosphorylation of P38 compared with ERK1/2 and JNK. Furthermore, when macrophages are stimulated by cytokines, ERK and JNK signaling pathways require more energy (such as, GTP) than P38 signaling pathways in response to progressive phosphorylation.

We thank you for your question that whether the inhibitors on these MAP kinases decrease pro-inflammatory cytokines and NO production. Affected by the epidemic COVID-19, our experiment has not been fully carried out for the time being. MAPK family members, which are proline-directed serine/threonine kinases, can be activated by upstream phosphorylation cascades such as MAP2K. The ERK1/2, JNK, and p38 members of MAPK may play vital roles in the generation of proinflammatory cytokines. According to our immunoblotting assay results, EP2 enhanced the expression of TNF-α, IL-1β, IL-6. So, we speculated that the inhibitors on these MAP kinases might decrease pro-inflammatory cytokines and NO production. In our future study, we will go into an in-depth study on the immune enhancement mechanism of EP2, including the roles of MAPK in the pro-inflammatory cytokines and NO production.

Comment 6: Did you check toxicity of EP2 on ICR mice? In Table 3, spleen index of EP2 80 mg/kg is heavier than that of CYP group, indicating that EP2 administration may cause enlarged spleen. Furthermore, EP2 stimulates RAW264.7 cells to produce pro-inflammatory cytokines and NO. Overproduction of these immunomodulatory substances may cause endotoxin-like shock in mice.

  Response 6: We are fully agreed with the reviewer’s comments. In our study, we checked the toxicity of EP2 on ICR mice. The mice received 80 mg/kg/d EP2 intraperitoneally for consecutive 14 days. The thymus index and spleen index were calculated as thymus and spleen weight. And we did not find EP2 showed toxicity on ICR mice. In Table 3, spleen index of EP2 80 mg/kg is heavier than that of control group. We speculated that the cyclophosphamide - induced immunosuppression mice might be more sensitive to EP2. And we found that EP2 at 40 mg/kg also could improve thymus index and spleen index, and showed no toxicity on ICR mice. In our furthermore, we will take a closer look at the drug concentration.

In our preliminary experiment of this study, we treated RAW 264.7 cells with EP2 for different times (0 h, 0.5 h, 1 h, 3 h, 6 h, 12 h, and 24 h). And we found that the storm of cytokines, iNOS and COX2 occurred after EP2 treated for 3 h or 6 h. And the levels of cytokines, iNOS and COX2 will be depressed when the cells were treated with EP2 for 12 h or 24 h. In this study, we aimed to study the proinflammatory effects of EP2 on RAW 264.7 cells, so we chose 5 time points (0 h, 0.5 h, 1 h, 3 h and 6 h) to detect the role of EP2. And the potent proinflammatory effects didn’t completely end for 24 h, but remained at a lower expression level. We speculated that the cytokines storm was triggered to promote cell resistance to foreign invasion when RAW 264.7 cells were treated with EP2 for 3-6 h. Then for the following period thereafter, the cells remained at a low level of cytokines and signaling pathway-related proteins, maintaining a "combat readiness" status, until the invader disappeared. EP2 didn’t cause overproduction of these immunomodulatory substances.

Comment 7: P1, L21, AMPK should be changed to MAPK.

Response 7: Thank you for your advice. We have corrected this mistake.

Comment 8: At Fig. 2A, presentation (Ctrl, 0.5…6) is not corresponding to bands, respectively. Correct it, please.  

Response 8: Thank you for pointing out this error. We have corrected this mistake.

Comment 9: P11, L328, RAW274.7 should be changed to RAW264.7.

Response 9: Thank you. We have revised.

Reviewer 3 Report

The present manuscript “Immunomodulatory effects of a low-molecular weight polysaccharide from E. prolifers on RAW 264.7 macrophages and cyclophosphamide-induced immunosuppression mouse models” described biological activities and medical application of EP1 and EP2 sulfated polysaccharide. The authors used 0.1 N HCl degradation, anionic exchange chromatography, autohydrolysis, and biogel P10 to separate, obtain EP1 and EP2 sulfated polysaccharide, especial immunological activity in EP2 part. One very recent publication in Int.J.Biol.Macromol (2020 May 1;150:1084-1092. doi: 10.1016/j.ijbiomac.2019.10.114. Epub 2019 Nov 20) also studied about structure and antioxidant effect.

Major Comments:

  1. The purity of EP1 and EP2: Do they still contain proteins/peptides or LPS contamination? Any scientific evidence?
  2. About lines 73-74: the sulfate content may also have some positive correlation with immune enhancement. Do authors have any evidence with or w/o sulfate of EP2, is the activity influenced?
  3. Usually, TLR4 and TLR-2 are receptors related to carbohydrate stimulation. Does authors assay the possibility of TLR-2 role in EP2 stimulation (lines 117-120)?
  4. Figure 4B-D: fold of content in Figure 4D comparing to western blot result, not identical. Also, Figure 5A, 5D, 5E: fold of content, not matched with western blot
  5. About animal study: IP injection, what is the absorption path of polysaccharide?
  6. About Table 1, UA uronic acid, how to determine glucuronic acid or galactocuronic acid or mannanouronic acid in EP2?

Minor Comments:

Figure 1. X-axis, working concentration ug/ml, and Figure 1C: EP1 and EP2 concentration

Figure 6: unit??

Correction:

  1. Table 1. SO4 into SO42-
  2. Figure 2A: time points, labeling position is shifting
  3. Figure 2H: D2, changes into EP2
  4. Line 134: reagent
  5. 8: RAW 264.7
  6. Line 151: reference # (Xing et al. 2015).
  7. Lines 272-274: why sediment 8g, the yield becomes, 20.1%?
  8. Line 275: 6 cm ×40 cm into 6 cm D x 40 cm H

Line 279: 2.6 × 100 cm into 2.6 cm D x 100 cm H

Line 287: 4.0 × 250 mm into 4 cm D x 25 cm H

Line 295: 4.6 × 250 mm into 4.6 mm D x 250 mm L

Line 296: 8.0×300 mm into 8.0mm D×300 mm

References:

2: Glycobiology into Glycobiol

3: 1091-1114 into 1091-114

5: 408-416 into 408-16

9: Int. Heart J. 2019, 60, 964-973

10: 611-9; 11: 1260-70; 15: 557-65; 16: 844-59; 18: 169-78

References 20, 22-28, 34, 35, 39, 40, 42

Author Response

Dear Reviewer,

I am very grateful to your comments for the manuscript in a timely and professional manner. We appreciate the concerns and suggestions provided, and have revised our manuscript accordingly. At this time, we have re-submitted the revised manuscript, and hope to have an opportunity to publish this paper in Marine Drugs. The revised manuscript with corrected sections marked in red is attached for easier editing and review purposes. The questions were answered below.

Comment 1: The purity of EP1 and EP2: Do they still contain proteins/peptides or LPS contamination? Any scientific evidence?

Response 1: Thank you. We applied a BCA protein assay kit (Beyotime Biotechnology, Shanghai, China) to detect the levels of proteins/peptides. We found that there was little proteins/peptides in EP2 (about 1.5%). The major compositions of the sulfated polysaccharides were total sugar and sulfate. There might be some proteins, ash content, moisture or others. In addition, the total sugar was determined by phenol sulfuric acid assay using rhamnose as a standard. Different monosaccharide had different absorbance in the same concentration, suggesting that there might be some errors for the total sugar analysis.

Before this study, we applied an endotoxin test as shown in Section 4.4. The concentration of endotoxin was determined by using the chromogenic end-point TAL assay kit (Solarbio, China). The experiment was performed according to the manufacturer’s instructions. The endotoxin in the sample activates a cascade of enzymes in TAL, the activated enzyme splits the synthetic substrate, releasing a yellow colored product with maximum absorbance at 405 nm. The yellow product can further react with diazo reagents forming purple product with maximum absorbance at 545nm. The absorbance of both yellow product and purple product are proportional to endotoxin levels. The polysaccharides were not contaminated with endotoxins, which indicate that the enhanced NO release was not induced by endotoxin compounds in the RAW 264.7 cells.

Comment 2: About lines 73-74: the sulfate content may also have some positive correlation with immune enhancement. Do authors have any evidence with or w/o sulfate of EP2, is the activity influenced?

Response 2: Thank you. In our preliminary experiment, we detected the immunomodulatory effects of EP2 and its degradations—DEP2, and we found DEP2 showed little immune-promoting effects on RAW 264.7 cells. DEP2 were the degradations of EP2, which were a series of mixtures of tetrasaccharides and hexosaccharides without sulfate groups. So, we speculated that the sulfate group played an important role in the immune-enhancement of EP2.

Comment 3: Usually, TLR4 and TLR-2 are receptors related to carbohydrate stimulation. Does authors assay the possibility of TLR-2 role in EP2 stimulation (lines 117-120)?

Response 3: TLRs family are the common receptors for inflammatory reactions. At present, there are many reports on the relation of TLR4 and the immunomodulation of sulfated seaweed polysaccharides. And in this study our focus was the effects of EP2 on pro-inflammatory cytokines and NLRP3 inflammasomes. Therefore, we selected TLR4 as the receptor to study the immunomodulatory effects of EP2 on RAW 264.7 macrophages. Now affected by the epidemic COVID-19, our experiment has not been fully carried out for the time being. In our future study, we will further study other receptors, including TLR2.

Comment 4: Figure 4B-D: fold of content in Figure 4D comparing to western blot result, not identical. Also, Figure 5A, 5D, 5E: fold of content, not matched with western blot

Response 4: Thank you. The analysis of protein expression in western blot was the mean value of three experiments. And we just put the most representative graph in the paper. Therefore, there were certain errors. However, the trends of protein changes presented by the data did not change.

Comment 5: About animal study: IP injection, what is the absorption path of polysaccharide?

Response 5: Thank you for your suggestion. The structure of polysaccharides is complex and the content of components is not single, so there are great problems in the study of the absorption. Because the polysaccharide has no light-emitting group, it cannot directly detect the absorption efficiency of polysaccharide after intraperitoneal injection. Some studies have used markers such as 5- dimethylaminonaphthalene-1-(N-(2- aminoethyl) sulfonamide (dansyl ethylenediamine,DED) and texas red (Texas Red) to label polysaccharides to trace their absorption. However, whether the polysaccharide is degraded and the degree of degradation in the process of absorption will have a certain effect on the absorption rate. In our lab, FITC labeled polysaccharides were used to track the absorption route in our previous research, but the labeling efficiency of FITC was too low. We are also currently looking for a more ideal method to study the absorption of polysaccharides.

Comment 6: About Table 1, UA uronic acid, how to determine glucuronic acid or galactocuronic acid or mannanouronic acid in EP2?

Response 6: According to the previous studies ([1] J.T. Kidgell, M. Magnusson, R. de Nys,C.R.K. Glasson, Ulvan: A systematic review of extraction, composition and function, Algal Research, 39 (2019) 101422. [2] Y. Yu, Y. Li, C. Du, H. Mou,P. Wang, Compositional and structural characteristics of sulfated polysaccharide from Enteromorpha prolifera, Carbohydr Polym, 165 (2017) 221-228. [3] Y. Li, W. Li, G. Zhang, X. Lu, H. Hwang, W.G. Aker, H. Guan,P. Wang, Purification and characterization of polysaccharides degradases produced by Alteromonas sp. A321, Int J Biol Macromol, 86 (2016) 96-104. [4] M. Lahaye,A. Robic, Structure and Functional Properties of Ulvan, a Polysaccharide from Green Seaweeds, Biomacromolecules, 8 (2007) 1765-1774.), there was no galactocuronic acid or mannanouronic acid in the polysaccharides from green algae. However, glucuronic acid and galactocuronic acid can be detected by the monosaccharide-PMP derivates HPLC method (J. Li, F. Gu, C. Cai, M. Hu, L. Fan, J. Hao,G. Yu, Purification, structural characterization, and immunomodulatory activity of the polysaccharides from Ganoderma lucidum, Int J Biol Macromol, 143 (2020) 806-813.)

Comment 7: Figure 1. X-axis, working concentration ug/ml, and Figure 1C: EP1 and EP2 concentration. Figure 6: unit??

Response 7: Thank you for pointing out this error. We have revised.

Comment 8: Table 1. SO4 into SO42-

Response 8: Thank you. We have revised as shown in Table 1.

Comment 9: Figure 2A: time points, labeling position is shifting. Figure 2H: D2, changes into EP2

Response 9: Thank you for pointing out this error. We have revised.

Comment 10: Line 134: reagent. 8: RAW 264.7. Line 151: reference # (Xing et al. 2015).

Response 10: Thank you for pointing out this error. We have revised.

Comment 11: Lines 272-274: why sediment 8g, the yield becomes, 20.1%?

Response 11: E. prolifera (1000 g) was extracted with 0.1 M HCl (30 L) at room temperature for 4 h. And then the supernatant was filtered through celite, neutralized, concentrated and precipitated by ethanol. The yield of the sediment was 20.1%. Then 8 g sediment underwent anion exchange chromatography on a DEAE-Bio Gel Agarose FF gel.

Comment 12: Line 275: 6 cm ×40 cm into 6 cm D x 40 cm H; Line 279: 2.6 × 100 cm into 2.6 cm D x 100 cm H; Line 287: 4.0 × 250 mm into 4 cm D x 25 cm H; Line 295: 4.6 × 250 mm into 4.6 mm D x 250 mm L; Line 296: 8.0×300 mm into 8.0mm D×300 mm

Response 12: Thank you for your suggestion. We have revised.

Comment 13: References: 2: Glycobiology into Glycobiol

Response 13: Thank you for pointing out this error. We have revised.

Comment 14: References: 3: 1091-1114 into 1091-114

Response 14: Thank you for your suggestion. We have revised.

Comment 15: References: 5: 408-416 into 408-16

 Response 15: Thank you. We have revised.

Comment 16: References: 9: Int. Heart J. 2019, 60, 964-973

Response 16: Thanks for pointing out this error. We have revised.

Comment 17: References: 10: 611-9; 11: 1260-70; 15: 557-65; 16: 844-59; 18: 169-78; References 20, 22-28, 34, 35, 39, 40, 42

 Response 17: Thank you for pointing out this error. We have revised.

Round 2

Reviewer 1 Report

The authors have made a number of textual changes to placate the reviewer, but failed to address the major problems with the study. Firstly, the authors provide no useful structural information about the sulphated polysaccharide which is teh key element in their study. They use analytical methods that they know provide incorrect results without any attempt at validation. They do not seem to care that the results that they are publishing are wrong. It is only in response to a reviewer's comments that they say that they knwo that their material is a mixture of at least two polysaccharides, one of which is reported to be biologically active. They do not put this in the paper, to hide that information from the general reader. How does the reader know that the reported biological results are not due to the second polysaccharide, where data has already been published?

Secondly, the authors use non-commutable colorimetric assays which, in my opinion are uncertain at the ±20%, and possibly ±50%, level, but they report results to three significant figure [reduced from four, to placate reviewer]. There does not seem to be a recognition that results should be reported with a precision compatible with the uncertainty.

In my opinion, papers of this type should include a credible structural characterisation of the molecule being tested: that is not true in this case.

Reviewer 2 Report

Dear authors

Thank you very much for your revised manuscript.

I would like to recommend  this manuscript to be acceptable in Marine Drugs.

Best regards

Reviewer 3 Report

the response 2 in the rebuttal letter can be added in the discussion or result part of the manuscript, in order to emphasize the essential biological role of the sulfate group in EP2.